# Unveiling the Machinery behind Chromosome Folding by Polymer Physics Modeling

**DOI:** 10.3390/ijms24043660

**Published:** 2023-02-11

**Authors:** Mattia Conte, Andrea Esposito, Francesca Vercellone, Alex Abraham, Simona Bianco

**Affiliations:** Dipartimento di Fisica, Università di Napoli Federico II, and INFN Napoli, Complesso Universitario di Monte Sant’Angelo, 80126 Naples, Italy

**Keywords:** chromatin architecture, polymer physics, loop-extrusion, phase-separation, gene regulation, epigenetics

## Abstract

Understanding the mechanisms underlying the complex 3D architecture of mammalian genomes poses, at a more fundamental level, the problem of how two or multiple genomic sites can establish physical contacts in the nucleus of the cells. Beyond stochastic and fleeting encounters related to the polymeric nature of chromatin, experiments have revealed specific, privileged patterns of interactions that suggest the existence of basic organizing principles of folding. In this review, we focus on two major and recently proposed physical processes of chromatin organization: loop-extrusion and polymer phase-separation, both supported by increasing experimental evidence. We discuss their implementation into polymer physics models, which we test against available single-cell super-resolution imaging data, showing that both mechanisms can cooperate to shape chromatin structure at the single-molecule level. Next, by exploiting the comprehension of the underlying molecular mechanisms, we illustrate how such polymer models can be used as powerful tools to make predictions in silico that can complement experiments in understanding genome folding. To this aim, we focus on recent key applications, such as the prediction of chromatin structure rearrangements upon disease-associated mutations and the identification of the putative chromatin organizing factors that orchestrate the specificity of DNA regulatory contacts genome-wide.

## 1. Introduction

Mammalian genomes are highly compartmentalized in the nucleus of cells [1,2,3,4,5,6,7], as revealed by independent sequencing-based technologies such as Hi-C [8], GAM [9,10] or SPRITE [11,12], and microscopy approaches [13,14,15,16,17,18,19,20,21,22,23,24]. From the gene up to the nucleus scale, chromosomes form a multilayered architecture of genomic interactions, comprising DNA loops [25,26], megabase-sized topological domains with enhanced levels of self-contacts (e.g., TADs) [27,28,29], higher-order structures (such as meta-TADs [30] and lamina-associated domains [31]), A/B compartments [8] and nuclear territories [32]. Such a complex three-dimensional (3D) organization has been increasingly linked to crucial functional roles, as, for example, DNA regulatory sequences (e.g., enhancers) can control the transcriptional output of distal genes by establishing specific, long-range contacts with their target promoters [1,33,34]. Those interactions are typically favored inside TADs, which are then thought to constrain the spatial communication between functional elements [2,3,35,36]. Disruption of TAD boundaries, indeed, has been shown to affect the network of gene-regulator contacts by altering the 3D organization of the genome, thus resulting in gene misexpression and disease [37,38,39]. On the other hand, recent advances in microscopy-based approaches, such as super-resolution multiplexed FISH techniques, pushed the investigation of those chromatin structures at the single-cell level [13,19,20,23]. Those studies, by allowing a direct visualization of chromatin conformations in individual nuclei, highlighted, for example, the abundance of TAD-like domains in single cells and their broad cell-to-cell structural heterogeneity, thus adding important constraints on chromosome folding beyond population-averaged contacts [13,40]. Additionally, recent live-cell imaging reports are enabling to track in time the dynamic of specific chromatin loci and the transient behavior of DNA functional interactions at the single-molecule level, unveiling the fourth dimension of genome topologies with a resolution of very few seconds [41,42]. 

However, despite those technological advancements, a clear understanding of the molecular mechanisms responsible for chromatin architectures is still lacking. In the last years, many different computational strategies have been developed to reproduce with high accuracy the complex patterns of chromosomal interactions as detected, e.g., in bulk contact data [43,44,45,46,47,48,49,50,51,52,53,54,55,56,57,58,59,60], albeit those methods provide only poor mechanistic insights on the processes underlying their origin and formation. To tackle such a major challenge, principled models from polymer physics, investigated within simplified yet rigorous theoretical frameworks, have been extensively used. In particular, two main classes of physics models, which rely on radically different biological scenarios of DNA contact formation, have been increasingly sustained by recent experimental evidence: loop-extrusion (LE) and polymer phase-separation [61,62,63,64,65,66,67,68,69,70,71,72,73,74,75,76,77,78,79,80,81,82,83,84,85,86,87,88,89,90,91,92,93,94]. 

Here, we review the key molecular ingredients of those different mechanisms of folding and describe their implementation into polymer physics models. We discuss their structural predictions at the single-molecule level that we test against recent super-resolution imaging data [13]. We show that both mechanisms are consistent with chromatin structure in single cells and that they can coexist to shape chromosome folding at the scale of TADs and functional interactions [64,65]. Next, we show that those polymer models can be employed to predict in silico the patterns of structural rearrangements upon disease-associated genomic mutations, clarifying how chromatin architecture impacts gene regulation in the case of specific structural variants [78,95]. Finally, we examine the molecular determinants of folding envisaged by the different models and show that, in the emerging picture, a specific combination of genome organizing factors, including but not limited to CTCF and cohesin, links 3D chromatin architecture to 1D chromatin states [95]. Overall, as they provide access to the underlying molecular processes, polymer physics models are shown to be crucial to start unveiling the complex machinery shaping chromatin spatial organization and valuable predictive tools for real-world applications, for example in biomedicine. 

## 2. Results

### 2.1. Loop-Extrusion and Polymer Phase-Separation as Mechanisms of DNA Contact Formation

In this section, we describe the basic ingredients of the LE and phase-separation models of chromosome folding. By using recent chromatin imaging data from multiplexed FISH experiments [13], we discuss how they perform in capturing the structural properties of specific human genomic loci at both the population-average and single-cell level, showing that they can cooperate simultaneously in single molecules to establish chromatin architecture. 

The LE envisages a biological scenario in which the spatial proximity between non-adjacent anchor sites is achieved by the translocation of SMC molecular complexes (e.g., cohesin rings) that extrude DNA loops along the chromatin chain, in a non-equilibrium process that requires energy burning, such as ATP hydrolysis (Figure 1a) [61,62,73,89]. The extrusion process halts in correspondence with specific blocking anchors, i.e., convergently-oriented CTCF binding sites, which mark the boundaries of consecutive genomic regions [61,62]. In different variants of the model, the extruding complexes are subject, for instance, to passive diffusion [91] or pushed, e.g., by transcription-induced supercoiling [84], or their positions along the chain are averaged out in order to build effective equilibrium models consistent with explicit-extruder approaches [96]. Polymer simulations based on LE have been used to explain, for example, the formation and compaction of mitotic chromosomes [97], organization of TADs in interphase [61,62], or the structural effects of CTCF/cohesin degradation at the cell population-average level [73,98]. Furthermore, recent single-molecule imaging experiments, although in simplified conditions, provided direct evidence of a cohesin or condensin-based extrusion activity in vitro [99,100,101,102]. Yet, besides those observations, other reports highlighted the role of additional chromatin architecture factors, including but not limited to CTCF and cohesin [103,104], in shaping chromosome structure. Those studies showed, for instance, that loop-extrusion is not essential for establishing functional (e.g., enhancer–promoter) interactions and controlling gene regulation [105,106,107,108], hinting that chromatin contacts can arise from distinct mechanisms. 

A different class of polymer models, exemplified, e.g., by the Strings and Binders (SBS) model [75,77], which relies on phase-separation mechanisms, represents the biological scenario whereby diffusing binders can drive the formation of a chromatin loop by bridging distal cognate DNA binding sites (Figure 1b, distinct types of binding sites are visually represented by different colors). The binders can represent single multivalent proteins, such as Transcription Factors (TFs), or, more generally, aggregates of several different molecules. In this framework, homotypic DNA interactions drive the spontaneous self-assembly of the system into specific globular domains, each associated with the local abundance of cognate binding sites [64,79]. The folding process in this case does not involve external energy input as it is sustained by the thermal bath. Such a mechanism is typically referred to as polymer–polymer (or chromatin-driven) phase-separation, as chromatin-associated proteins can induce the formation of collapsed polymer globules by cross-linking different cognate DNA segments [109]. This class of models can also include DNA binding molecules which have specific multivalent interactions with each other: in this case, the system undergoes a different folding process, known as liquid–liquid phase-separation, in which the interactions among soluble molecules promote the assembly of liquid-like protein droplets around the corresponding binding sites on chromatin [109]. Those nuclear droplets can also be stable in the absence of the polymer scaffold and are consistent with those observed in recent experiments, such as combinations of Pol-II, TFs, nuclear bodies, and coactivators linked to transcriptional control [110,111,112,113,114,115,116]. This scenario can be theoretically investigated, e.g., in SBS-like polymer models by allowing specific binder–binder interactions that could mimic, for example, those observed between many proteins, chromatin modulators, and RNA molecules in the cell [117,118,119]. In a more refined model, DNA-binding proteins could also switch between active and inactive binding states, providing a theoretical system to explore the possible role of a time-dependent switch of chromatin epigenetics on folding [120]. Polymer models that investigate the interplay between the formation of liquid-like protein aggregates and protein-DNA binding in shaping chromatin architecture are extensively discussed, e.g., in [109,121,122]. Notably, these models predict that even when a direct mutual interaction of the binders is not included, the binders can cluster and form phase-separated molecular aggregates under suitable conditions, via a process named bridging-induced attraction, recently observed also by microscopy experiments [82,108,123,124]. Finally, a model with direct polymer-to-polymer attractive interactions (that mimic, e.g., internucleosomal interactions between histone molecules bound to DNA), rather than protein-mediated, reproduces similar behaviors as dictated by polymer thermodynamics [65,125]. 

Different, and complementary, strategies have been developed to identify the putative binding sites of phase-separation-based polymer models, which mainly differ in using or not a priori knowledge of biological information, such as epigenetic marks and binding molecules. The first strategy, in which the model binding sites are derived, e.g., from epigenetics, is useful to test the molecular hypothesis, albeit by definition it cannot identify novel, additional molecular factors beyond those used to inform the model [60,63,68,93]. In the second approach, the binding domains are inferred with no prior epigenetics information and derived, e.g., from available contact data only. Such a strategy is exemplified, for instance, by the polymer-based recursive statistical computational procedure (PRISMR) [78], which infers the optimal SBS polymer model to best-fit input pairwise contact data (e.g., bulk Hi-C or GAM) [95,126]. Supported by increasing experimental evidence that traces back to phase-separation mechanisms [108,110,112], those models provided, overall, a quantitative explanation of contact formation across genomic scales, from the sub-TAD scale up to nuclear compartments [67,71,73,77,79,81,93,94], and also at the single-cell level [64]. To set a reference within such a broad spectrum of models, we consider hereafter the SBS as a representative phase-separation-based polymer model. 

The technical implementation of those distinct folding mechanisms (loop-extrusion and phase-separation) into polymer models is broadly discussed in specialized papers (see, e.g., [61,64,73,79,127]) and for brevity will not be addressed here. Briefly, in those models, chromatin is represented as a coarse-grained polymer chain of consecutive, non-overlapping beads subject to classic physical potentials (i.e., FENE or harmonic and properly truncated Lennard-Jones functions) and regulated by a stochastic Langevin dynamics investigated by massive Molecular Dynamics (MD) simulations in the steady-state [128,129,130]. In the case of the LE model, the extruders stochastically bind to the polymer binding sites and translocate on them until they encounter a convergently-oriented pair of blocking sites or another extruder, or they stochastically dissociate from the chain. In the SBS model, the binders, initially randomly located in the system, establish homotypic attractive interactions with their cognate sites, hence driving a phase transition of the polymer into an equilibrium phase-separated state in which the chain is partitioned into specific and spatially segregated globular domains. MD simulations are typically performed in cubic boxes with periodic boundary conditions and optimized by using a high-performance toolkit, such as OpenMM [131], LAMMPS [132], or HOOMD [133].

#### 2.1.1. The Polymer Models Are Consistent with the Structure of Chromatin at the Population-Average Level

To illustrate the predictive power of the models in explaining chromatin conformations, we consider as a case study a 2.5 Mb wide genomic region (Chr21: 34.6–37.1 Mb, hg38) in a human colon cancer cell line (HCT116) where single-cell microscopy data are available [13] (Figure 2, top panel). 

In the LE model of the locus (Figure 2, middle), the blocking anchor sites correspond to the genomic locations of CTCF ChIP-seq peaks available in the studied chromatin region from the ENCODE database [134]. Furthermore, to consider the epigenetic diversity of single-cell landscapes, those anchor sites are present in each single-polymer molecule with a specific probability value (roughly in the range of 20–80%), chosen to best match the average (e.g., Hi-C) contact data of the locus [65,81]. Similarly, since a principled approach to parameterize the model is missing [73], other LE parameters, e.g., the processivity of the cohesin rings along the chain and their number, are selected upon a systematic parameter sweep suited to best reproduce average contact or distance data [65,73]. In the studied locus, the processivity, that is the extrusion velocity divided by the extruder dissociation rate, is 700 kb, while 10 is the number of extruding factors [65]. We point out, however, that fresh data from live-cell imaging technologies could help in a more precise and quantitative calibration of the LE dynamic parameters, as also discussed in recent studies [41,42]. 

On the other hand, the SBS model of the locus (Figure 2, bottom) has four distinct types of binding domains (visually represented by different colors), which are associated with the major TAD-like domains visible in the experimental median distance map of the locus. Each type of binding site (i.e., each color) of the model has been shown to correlate significantly with a specific, distinct combination of epigenetic factors (e.g., H3K4me3, H3K27ac, H3K27me3) or architectural proteins (such as CTCF/Cohesin and Pol-II) [64], hinting that a combinatorial action of different molecular factors shapes the polymer phase-separation of the studied chromatin region [64,95]. The energy affinities between the binders and the binding sites of the SBS chain are set in the weak biochemical energy scale (i.e., 1–10 K_B_T, K_B_ is the Boltzmann constant and T the temperature), while binder concentrations are in the range of tens nmol/l (i.e., 50–500 nmol/L) [64,79]. Within such a robust range of energy/concentration parameters, the SBS chain undergoes a thermodynamic phase transition from an initial coil (i.e., randomly folded) conformation to an equilibrium globule phase-separated state (where the polymer self-assembles into segregated micro-globular structures each enriched of cognate binding sites) [64]. 

As a first benchmark of the LE and SBS models, their median spatial distance matrix (i.e., the median of the distance maps of each single-polymer conformation) has been compared against the corresponding locus map from imaging data [65]. It was found that both models faithfully recapitulate the ensemble-average conformation of the HCT116 locus, as they capture its main TAD structures and loop interactions (the latter particularly enriched upstream, e.g., in correspondence of the *RUNX1* and *SETD4* genes, Figure 2). The high degree of similarity between the model and experimental distance matrices is quantified, e.g., by their high distance-corrected Pearson correlation coefficients [78]: r’ = 0.87 for LE and r’ = 0.84 for SBS (Figure 2). As a further check of the significance of the results, a null control model, for instance, a self-avoiding chain with no mechanism in place, has been shown to return only poorer r’ correlations, i.e., roughly 7 times lower [135]. Similar findings on the performance of the models have been also reported in other genomic loci and in different cell lines, e.g., in human fibroblast cells where super-resolution microscopy data are available [13,19,64,65]. 

Taken together, those results highlight that active processes, such as DNA loop-extrusion, and passive mechanisms, such as polymer phase-separation, both work well in recapitulating the structural properties of chromatin architecture, e.g., TADs and loops, at the cell population level.

#### 2.1.2. Loop-Extrusion and Polymer Phase-Separation Cooperate in Determining Heterogeneity of Chromatin Architectures in Single Cells

Next, we aim to discuss the loop-extrusion and phase-separation scenarios of folding at the single-cell level. To this aim, the statistical ensemble of single-polymer conformations predicted by the LE and SBS models [65] was compared against the imaged single chromatin structures of the locus in HCT116 cells (Figure 3a) [13]. To address whether the different models do provide a statistically significant, bonafide, representation of the microscopy conformations, the root-mean-square deviation (RMSD) criterion was used to associate each experimental 3D structure to a corresponding best-match single-polymer conformation of the models [65,136]. In brief, by performing a roto-translational alignment of model and microscopy structures to optimize the RMSD of their spatial coordinates, the criterion identifies, for each imaged single-cell structure, the corresponding model conformation with the least RMSD [65,137]. A visual example of the RMSD best-match procedure is reported in Figure 3b (top panel): a considered microscopy structure (cell 75 is shown as an example) has a single-cell distance matrix with two, spatially separated, TAD-like domains, reflected in two main segregated globules in 3D space; such a specific topology is steadily found in the corresponding best-matches predicted, respectively, by the LE and SBS models (Figure 3b, top panel), highlighting, at least visually, the consistency of the method. To properly assess the significance of the RMSD procedure, the distribution of RMSD values between random pairs of microscopy conformations was set as a control [64]. It was found that the RMSD distribution of the best-matching experiment–model pairs only poorly overlaps the control (two-sided Mann–Whitney *p*-value < 10^−3^, Figure 3b, middle), with more than 90% of the LE and SBS best matches falling below 10% of control (pie charts in Figure 3b, bottom). Similar results are found by taking control polymer models without LE- or SBS-based mechanisms (e.g., chains with only self-avoidance effects) [138]. 

Overall, the structural comparison via the RMSD criterion shows that the single-molecule conformations predicted by loop-extrusion and polymer phase-separation are statistically consistent with chromatin imaged structures at the single-cell level [65]. 

Finally, in order to understand whether the different models could provide a theoretical framework to explain the origin of the microscopy-reported cell-to-cell variability of the locus [13], the level of structural heterogeneity in imaging data has been quantified by computing the distribution of r’ correlations between all pairs of experimental single-cell distance matrices [64,139]. Consistent with the substantial variability of the observed single-locus conformations [13], the experimental distribution has a broad variance (Var(r’) = 0.20, Figure 3c, blue distribution) that is overall well reproduced by the LE and SBS models (respectively, in orange and green). However, the non-zero average value of the experimental distribution (r’ = 0.27, vertical dashed line in Figure 3c) signals that, albeit broadly varying, the imaged conformations have a residual structural correlation, which is, respectively, under- and over-estimated by models relying solely on loop-extrusion or phase-separation [65] (Figure 3c). Those small discrepancies could be related to the different nature of the underlying LE or SBS processes, as, for instance, the first tends to favor the formation of strong, yet more fleeting (i.e., less correlated) loop contacts, whereas the second tends to establish spatially segregated, stable (i.e., more correlated) globular domains. Consistent with such a picture, a polymer model that combines simultaneously both mechanisms at the single-molecule level (LE + SBS, Figure 3c) returns an average value consistent with the data, indicating that loop-extrusion and phase-separation can indeed coexist to establish chromatin architecture and shape its full range of heterogenous structural realizations in single cells [65]. 

Summarizing, those recent studies suggest that the observed cell-to-cell variability of chromatin conformations is consistent with a more complex scenario of folding in which loop-extrusion and polymer phase-separation cooperatively act at the single-molecule level to shape genome structure. Furthermore, as different phase-separation mechanisms may take place in the nucleus, an interesting perspective would be to dissect their specific roles in chromatin folding, for instance by considering more sophisticated models with a prominent liquid phase along with a chromatin-driven phase and active LE. The exact interplay and dynamics of this cooperation are yet to be comprehensively explored and could be one of the grand challenges of the field for both experimental and theoretical tests.

### 2.2. Further Validation and Key Applications of Chromatin Polymer Models

A central role for validated models from polymer physics is their ability to make accurate predictions on chromatin architecture that can be tested against independent experiments.

In this respect, applications of polymer models have been crucial, for example, in understanding the role of chromatin architecture in human genetic diseases. Recent studies have shown, in particular, that the SBS model can accurately predict the effects on chromatin 3D structure of disease-associated genomic structural variants (SVs), such as deletions, inversions, and duplications, thus helping to understand their contribution in driving disease [78,95,140,141,142,143]. To illustrate this point, in the following subsection, we focus on recent applications at a crucial chromatin locus, *Sox9*, linked to embryonic development [95]. On the other hand, while the phase-separation-based SBS model can well explain the effects of large mutations such as SVs, the LE model has been shown to be particularly suited to predict the effects of much smaller genomic rearrangements, such as single CTCF sites mutations [62]. Interestingly, however, an SBS polymer model including explicit interactions between preferential CTCF sites, as in the LE, has been shown to slightly improve predictions about the effects of SVs, indicating that the cooperation of the two mechanisms, consistent with the findings discussed above, can better explain chromatin folding also in disease [78].

Another relevant aspect of chromatin polymer models is that they are helping to spread light on the molecular factors that organize chromatin folding across chromosomal scales. Indeed, different studies based on polymer modeling and supported by experiments have revealed that the formation of different chromatin sub-compartments and specific folding patterns are driven, e.g., by homotypic interactions between chromatin sites sharing similar chromatin occupancy features, including histone modifications, active and poised gene promoter states, and TF binding sites [64,67,68,81,93,95]. To further elaborate on this point, we focus in the last subsection on a recent study that, by employing the SBS model, identifies the key putative chromatin organizing factors that are involved in the specificity of DNA regulatory contacts genome-wide [95].

#### 2.2.1. Prediction of Genomic Mutations Effects

To exemplify the ability of the SBS model to predict the effects of disease-associated mutations on chromosome conformation, we take here as a case study the developmental-related *Sox9* locus, in which different and partially overlapping SVs were shown to produce different phenotypes [144]. The SBS model specific to the locus has been built based on wild-type (WT) cHi-C data in mouse E12.5 limb buds (Figure 4a) by using the PRISMR approach [78,95], which infers the SBS model that best describes the locus contact matrix, i.e., the minimal number and positioning of the binding site types in order to obtain an ensemble of polymer 3D structures consistent with input data. The SVs are implemented in silico on the WT SBS model and their corresponding contact matrices, derived from polymer physics with no fitting parameters whatsoever, are compared to independent cHi-C data from cells that carry those precise mutations [95]. 

As exemplificative cases, in Figure 4b we show the result of a 0.4 Mb duplication (*Dup-S*) in the non-coding DNA region within the *Sox9* gene TAD (intra-TAD duplication) associated with female-to-male sex reversal in humans, whereas in Figure 4c we illustrate the result for a 1.6 Mb duplication (*Dup-L*) encompassing the neighboring TAD boundary (inter-TAD duplication) with no phenotypic effects. The Pearson and distance-corrected Pearson coefficients between the model-predicted and cHi-C contact matrices reflect their good degree of similarity being, respectively, r = 0.95 and r’ = 0.76 in *Dup-S* and r = 0.92 and r’ = 0.63 in *Dup-L*. Specifically, and consistent with experimental data, the model predicts that while the intra-TAD duplication *Dup-S* does not affect the overall TAD structure, the inter-TAD duplication *Dup-L* produces the formation of a separate chromatin domain termed “neo-TAD” [144]. However, the neo-TAD results are spatially insulated from its neighboring regions thus preventing ectopic interactions between *Sox9* and its regulatory sequence, explaining the lack of pathogenicity in humans carrying this type of mutation.

The polymer model of the *Sox9* locus allows dissecting of the interactions of the duplicated from the original DNA sequence, information not accessible through only Hi-C data [95]. Importantly, while *DupS* is fully included within the TAD encompassing *Sox9*, in *DupL* the duplicated region spans two TADs (Figure 4b,c). Since a TAD, in the model, derives from the presence of a dominant type of binding sites in that region (see, for example, Figure 2, bottom), the duplicated and original sequence in *DupS* share many homologous binding sites producing an excess of interactions localized around the mutated region, without altering the overall contact pattern. The 3D conformation of the *Sox9 DupS* locus (Figure 4b) shows indeed that the duplicated region (colored in red in the figure) remains well embedded into the original sequence (in brown). On the other hand, in the inter-TAD duplication *DupL*, since different prevailing binding sites are present, the portion of the duplication within the *Sox9* TAD interacts with itself more than with its neighboring regions, remaining partially isolated from the rest of the locus. The snapshot of the 3D structure of *DupL* locus shows, indeed, that the region corresponding to the neo-TAD (in red) is spatially isolated.

Summarizing, in addition to demonstrating that the SBS model can be successfully used to predict the effect of disease-associated mutations on genome 3D structure, those results clarify how mutations differently impact chromatin architecture and gene regulation, resulting in distinct phenotypes.

#### 2.2.2. A Combinatorial Epigenetic Code Linking 1D and 3D Chromatin Organization

The genomic size of the chromatin regions discussed so far is in the range of a few millions of base pairs. However, the folding machinery also acts at much larger scales, up to the size of whole chromosomes [7,30]. Chromosome-wide polymer models are then fundamental to test the validity of the underlying physical mechanisms. Here, we show that the SBS model can be successfully employed genome-wide and discuss how that allows to both identify the location and combination of the putative binding sites underlying chromatin contacts and to derive a first characterization of their molecular features.

The ability of the SBS model in describing the folding at larger genomic length scales has been tested, e.g., by using high resolution (5 kb) in situ Hi-C data in the human lymphoblastoid cell line GM12878 [25]. For each chromosome, PRISMR [78,95] has been used to infer the SBS model that best describes its corresponding contact matrix. Importantly, PRISMR exploits only Hi-C data as input, with no prior knowledge of binding factors. As an example, Figure 5a shows the results of the PRISMR procedure for chromosome 20, where experimental and model contact patterns show a quantitative degree of similarity (r’ = 0.85). The position and abundance of the different types of binding sites (binding domains) are also shown (Figure 5a, middle panel). Interestingly, the arrangement of the binding site types along the polymer chain is highly non-trivial: rather than being restricted to small, contiguous genomic regions (as found, e.g., in common and much simpler block-copolymer models), the binding domains do overlap with each other and are spread across several mega-bases, hence capturing contacts occurring up to the chromosomal scale [95]. These results indicate that the basic molecular ingredients considered by the model are sufficient to explain contact patterns across genomic scales.

Since the model uses as input only the GM12878 Hi-C contact matrices, its inferred binding sites have been characterized a posteriori, by crossing their position along the genome with independent biological information. In particular, five key histone marks available from the ENCODE database in the same cell line [134] have been considered and the correlation between their genomic signals and the genomic location of the model binding domains computed, thus obtaining for each domain its epigenetic signature. By grouping the domain histone profiles across chromosomes via hierarchical clustering algorithms, it has been discovered that there are 9 statistically different groups (or epigenetic classes) [95]. Figure 5b shows the average histone profile per class. Interestingly, each class correlates with a specific combination of different epigenetic factors rather than with a single one. For instance, although there are three classes that strongly correlate with active chromatin marks (Active 1, Active 2, and Active 3 in Figure 5b), class 1 is enriched for only active marks, while classes 2 and 3 are both enriched in H3K9me3 (also found in heterochromatin) and class 3, in particular, shows a stronger correlation with H3K4me1, a histone mark associated especially with active enhancer regions [145]. That suggests that the binding domains corresponding to the first three classes produce contacts between transcribed and regulatory regions. Importantly, the first six classes correlate with CTCF (Figure 5b), confirming its significance in regulating chromatin architecture and gene activity [146], but also indicating that its role can be modulated by different sets of histone marks and molecular factors as discussed above.

Summarizing, the binding domains inferred genome-wide have a specific barcode of histone marks and fall in epigenetic classes that match well the chromatin states found by previous segmentation studies [14,145,147,148,149]. However, while the latter are, by definition, non-overlapping segments, the binding domains do overlap with each other along the genome, providing each DNA window with a distinctive set of binding site types that can be read as a combinatorial code linking 1D and 3D chromatin organization [95].

## 3. Discussion

In this work, we discussed principled models of polymer physics to explore the complex machinery underlying chromosome spatial organization. To this aim, we reviewed two major classes of models, i.e., loop-extrusion and polymer phase-separation, that envisage distinct biological pictures of folding. In the first class, chromatin structures, e.g., loops and TADs, are established by molecular motors (such as SMC proteins) that extrude DNA loops in an off-equilibrium, cell-energy-consuming, process [61,73,89]. In the second, those topological structures are spontaneously shaped by molecular diffusing binders (corresponding to specific combinations of chromatin organizing factors, such as CTCF/cohesin, histone marks, TFs, active/poised Pol-II) that bridge distal DNA cognate sites via equilibrium polymer thermodynamics mechanisms [64,77,79]. The phase behaviors described in such models are typical of complex systems of physics [150,151,152,153,154]. In the emerging scenario, consistent with single-cell imaging data, chromatin contacts at the megabase-scale arise from the interplay of those distinct (respectively, active and passive) physical processes, which can coexist and act simultaneously at the single-molecule level [65]. 

The models we considered, albeit simplified, enable quantitative predictions on real chromatin structures that can be tested against independent experiments. For example, within the framework of the SBS (i.e., phase-separation-based) model, we discussed how specific genomic mutations at key chromatin loci (such as the *Sox9* locus) perturb 3D chromatin structure, producing ectopic contacts consistent with real data (e.g., via specific enhancer-hijacking events) that result in distinct phenotypes [95]. Importantly, such predictions can help the interpretation of human genetic variants in driving severe diseases, such as congenital disorders and cancers, which is currently a challenging question in many clinical cases. Recently, and interestingly, those validated models have been also shown, in a reverse approach, to serve as ground truth to benchmark independent experiments, for instance by assessing advantages and limitations of technologies for the determination of chromosomal structure [138,155], therefore expanding the spectrum of their useful applications in silico.

Nevertheless, the search for the different mechanisms controlling chromosome architecture is still open and debated, and many long-standing questions of DNA biology, such as the molecular rules shaping enhancer–promoter communication, are only poorly addressed from first principles. In this respect, a few models of long-range transcriptional control have been recently proposed, invoking, e.g., longer-lived promoter states [36], “activity-by-contact” models relying on chromatin state measurements [156], or TF-grounded-activity models based on local 3D gradients of chemical signals [157], yet a comprehensive understanding at the molecular level is yet far from being reached. For that reason, validated theories from polymer physics that rely on robust organizing principles can be crucial to elucidate the relationship between genome architecture and function, which we expect to be among the most critical challenges of the field in the nearest future. 

## Figures and Tables

**Figure 1 ijms-24-03660-f001:**
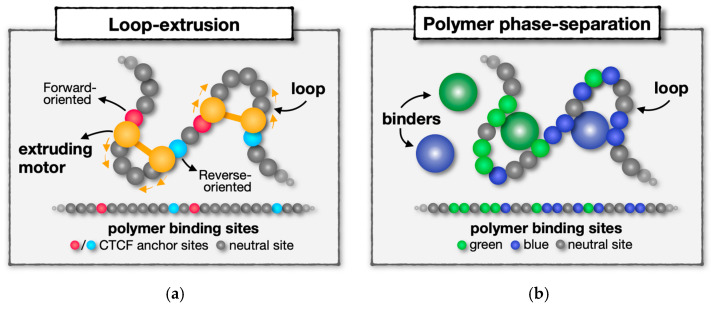
Two main distinct physical mechanisms of DNA contact formation. (**a**) Cartoon of the loop-extrusion model: active motors (e.g., cohesin rings, depicted in orange in the figure) translocate along the chromosome and extrude DNA loops, hence defining a physical contact between distal, blocking, anchor sites (e.g., convergently-oriented CTCF sites, in red and light blue) where translocation halts. (**b**) Cartoon of phase-separation-based polymer models: diffusing binders, such as TFs, enable spatial proximity by bridging pairs (or multiplets) of distal cognate sites (each type represented by a different specific color), thus driving the spontaneous self-assembly of the chain into phase-separated clusters of sites with enriched levels of self-interactions.

**Figure 2 ijms-24-03660-f002:**
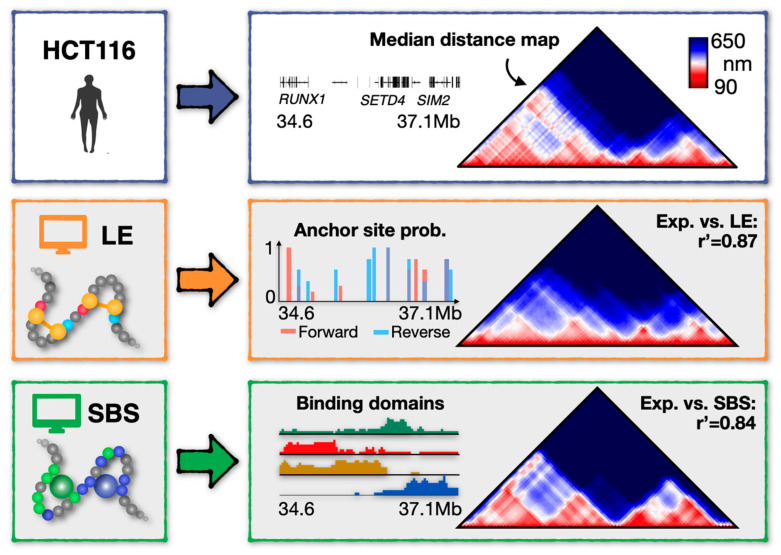
Loop extrusion (LE) and Strings and Binders (SBS) models of a 2.5 Mb wide chromatin locus in human HCT116. The experimental median distance matrix of the locus (top) [13] is consistently recapitulated by the LE (middle) and SBS (bottom) models, as highlighted by their high distance-corrected correlation coefficients (respectively, r’ = 0.87 and r’ = 0.84), indicating that both mechanisms significantly capture the spatial conformation of the locus at the ensemble-average level. Adapted from [64,65].

**Figure 3 ijms-24-03660-f003:**
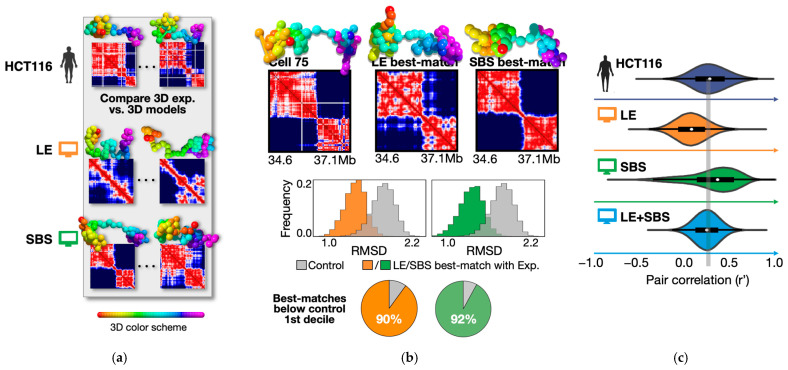
LE and SBS cooperate in determining the heterogeneity of chromatin architectures in single cells. (**a**) Schematic of the all-against-all structural comparison between microscopy [13] and model [65] 3D structures. (**b**) Top: example of experiment–model best-match association via the RMSD criterion. Middle: the identified RMSD best matches are statistically significant in the LE and SBS models (two-sided Mann–Whitney *p*-value < 10^−3^, RMSD are z-scored on the *x*-axis). The control is the distribution of RMSD values for randomly taken pairs of experimental conformations. Bottom: more than 90% of the model best matches fall below the first decile of the control distribution. (**c**) The degree of cell-to-cell structural variability is quantified by an all-against-all pair correlation of single-molecule distance matrices. While LE (orange distribution) and SBS (green), respectively, under- and over-estimate the experimental distribution (blue) of pair correlations, a model combining both processes at the single-cell level (LE + SBS, light blue) best explains the data, hinting that LE and SBS cooperate in determining chromatin structure and its observed cell-to-cell heterogeneity. The dashed vertical line is the average value of the experimental distribution. Adapted from [64,65].

**Figure 4 ijms-24-03660-f004:**
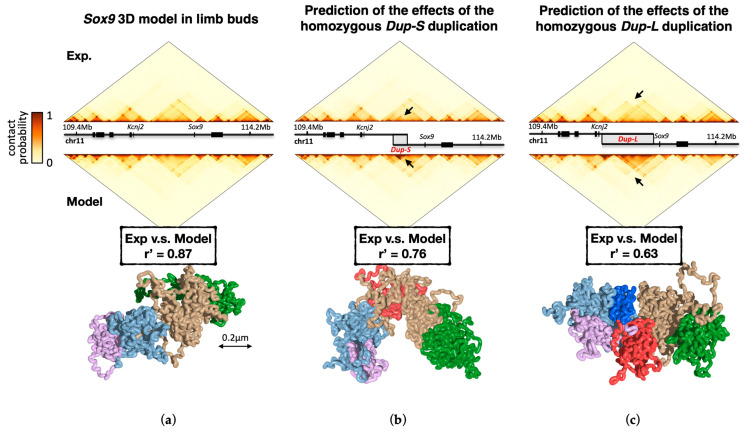
Prediction of the impact of genomic mutations. (**a**) Hi-C contact matrix [144] of the wild-type (WT) *Sox9* genomic region (**top**) and its corresponding SBS model (**middle**) show high similarity (r’ = 0.87). **Bottom**: snapshot of a single-molecule conformation in the WT predicted by the model. (**b**) The model contact matrix (**middle**) in the *Dup-S* case, predicted by implementing the duplication on the WT model. It shows a high correlation (r’ = 0.76) with the corresponding matrix derived from an independent experiment (**top**) in cells bearing the *Dup-S* mutation. The black arrows highlight the effect of the duplication on the contact pattern. **Bottom**: snapshot of a model-predicted single-molecule conformation in *Dup-S* (note, in particular, how the duplicated region, highlighted in red, is well embedded into the original sequence, colored in brown). (**c**) The model contact matrix (**middle**) in the *Dup-L* case, predicted by implementing the duplication on the WT model. It shows a correlation with a matrix from an independent experiment (**top**) in cells bearing the *Dup-L* mutation as high as r’ = 0.63. The black arrows highlight the effect of the duplication on the contact pattern. Differently from the *Dup-S* case, a snapshot of the 3D structure in *Dup-L* (**bottom**) shows that a fraction of the duplicated region (neo-TAD, colored in red) results to be spatially isolated. This clarifies that different mutations result in different 3D structures and distinct enhancer-hijackings, explaining their phenotypes. Adapted from [95].

**Figure 5 ijms-24-03660-f005:**
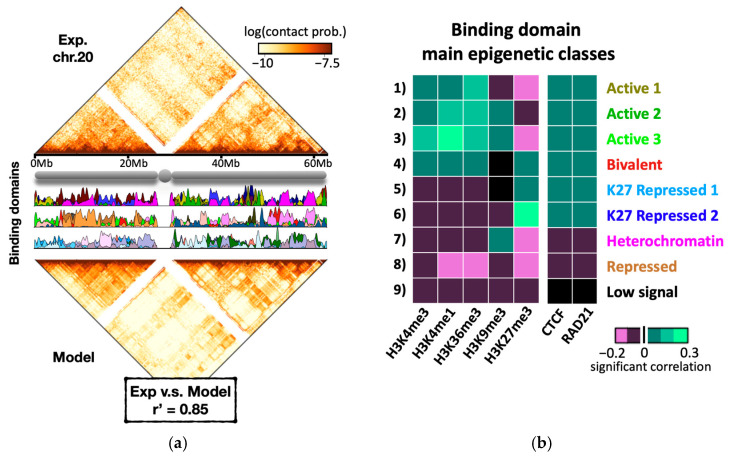
A combinatorial epigenetic code links 1D and 3D chromatin structures. (**a**) Top: in situ Hi-C data [25] of chromosome 20 at 5 kb of resolution in the human GM12878 cell line. Middle: position and abundance of the different binding domains along chromosome 20. The colors reflect the specific association of each domain with a set of key epigenetic marks (see panel **b**). Bottom: the contact matrix inferred from the model shows a contact pattern similar to Hi-C (r’ = 0.85). Similar results are found across chromosomes [95]. (**b**) According to their correlation with key histone marks, the genome-wide binding domains of the model cluster in nine epigenetic classes matching well the chromatin states found in epigenetic segmentation studies [95]. The heatmap shows the histone profile of the centroid of each class. Interestingly, some but not all the classes also correlate with CTCF and cohesin, suggesting that a specific combination of genome organizing factors, including but not limited to CTCF and cohesin, links 3D chromatin architecture to 1D chromatin states. Adapted from [95].

## Data Availability

Data are contained within the article.

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
