# Peer review of "Unveiling the Machinery behind Chromosome Folding by Polymer Physics Modeling"

_ijms, 2023, doi:10.3390/ijms24043660_

Round 1
Reviewer 1 Report
The review treats the topic of the driving principles of chromatin folding in cell nucleus, a long standing endeavor of the interdisciplinary community of molecular biologists and biophysicists, and yet a very central topic of the discussion of the field.
After a brief introduction of the topic two main models are discussed in detail, loop extrusion and phase separation (LE and PS), claiming that these are the two main mechanisms explored by the field and with a good degree of comparison with experiments. I agree with this statement, and I appreciate the effort by the authors to cover different aspects of phase separation that is indeed quite a broad spectrum of models.
For instance, the citation of the work by Erdel&Rippe 2018 is quite appropriate, even if I would debate more the differences between a pure liquid PS and the chromatin-driven PS that the authors dissect in detail from section 2.1.1 onwards, and I am interested in the author’s view of considering models with a prominent liquid phase along with a chromatin-driven phase and LE.
The exposition of the SBS method and its extension to LE is adequate and well connected with the real experiments. The single cell analysis reported is very insightful for the reader. The section on the structural variants is an important example of the powerful predictions that such quantitative models, like polymer physics models, can produce.
The discussion correctly admits that we are still far from a complete understanding of the organizational principles and the role of the biochemical factors involved, citing some recently proposed models. I believe it would be good to briefly discuss the possible role of time-dependent epigenetic heterogeneity on chromatin folding, for instance citing the model of Cook&Marenduzzo (BioPhys J 2017) that hint at the importance of the time switch of chromatin epigenetics to preserve its steady state architecture, since it is a poorly studied argument both theoretically and experimentally (for technical difficulties) but that nevertheless is arguably present and might affect our ability to correctly predict chromatin architecture.
I therefore approve the manuscript for submission when the two papers I have previously cited will be briefly discussed as explained above, or the authors explain why not to do so.
Author Response
We thank the Reviewer for this valuable advice and have revised our text accordingly. As suggested, to further debate the differences between a pure liquid- and chromatin-driven phase separation, we have extended Section 2.1 of the manuscript and added new references (Banani et al. Nat Rev Mol Cell Biol 2017; Brangwynne et al. Nat Phys 2015; Hnisz et al. Cell 2017). We have also added a reference to Brackley et al. BioPhys J 2017 in the same section to highlight the importance of protein switching and time-dependent epigenetic heterogeneity on chromatin organization. Finally, we appreciate the Reviewer’s intuition of considering a model with a prominent liquid phase along with a chromatin-driven phase and LE and have added new sentences in Section 2.1.2.to comment on such an interesting extension of the considered models.
Reviewer 2 Report
In my opinion, the authors have analyzed a large body of material on the organization of chromatin. The authors consider folding mechanisms from the point of view of polymer physics models. The development of this approach will make it possible to predict the structural rearrangements of the genome, including pathology. I think that the article can be accepted for publication.
Author Response
We thank the Reviewer for the positive comment and appreciation of our work.